# Genetic Mapping of Behavioral Traits Using the Collaborative Cross Resource

**DOI:** 10.3390/ijms24010682

**Published:** 2022-12-30

**Authors:** Wei Xuan, Ling Zhang, Yu Zhang, Xiuping Sun, Jue Wang, Xianglei Li, Lingyan Zhang, Xinpei Wang, Grant Morahan, Chuan Qin

**Affiliations:** 1NHC Key Laboratory of Human Disease Comparative Medicine, Beijing Engineering Research Center for Experimental Animal Models of Human Critical Diseases, International Center for Technology and Innovation of Animal Model, Institute of Laboratory Animal Sciences, Chinese Academy of Medical Sciences (CAMS) & Comparative Medicine Center, Peking Union Medical College (PUMC), Beijing 100021, China; 2Harry Perkins Institute of Medical Research, QEII Medical Centre and Centre for Medical Research, The University of Western Australia, Nedlands, Perth, WA 6009, Australia; 3Changping National Laboratory (CPNL), Beijing 102200, China

**Keywords:** collaborative cross mice, behavioral genetics, QTL mapping, high precision

## Abstract

The complicated interactions between genetic background, environment and lifestyle factors make it difficult to study the genetic basis of complex phenotypes, such as cognition and anxiety levels, in humans. However, environmental and other factors can be tightly controlled in mouse studies. The Collaborative Cross (CC) is a mouse genetic reference population whose common genetic and phenotypic diversity is on par with that of humans. Therefore, we leveraged the power of the CC to assess 52 behavioral measures associated with locomotor activity, anxiety level, learning and memory. This is the first application of the CC in novel object recognition tests, Morris water maze tasks, and fear conditioning tests. We found substantial continuous behavioral variations across the CC strains tested, and mapped six quantitative trait loci (QTLs) which influenced these traits, defining candidate genetic variants underlying these QTLs. Overall, our findings highlight the potential of the CC population in behavioral genetic research, while the identified genomic loci and genes driving the variation of relevant behavioral traits provide a foundation for further studies.

## 1. Introduction

Genetic studies of behavioral diversity demonstrated heritable variation for traits such as locomotor activities, cognition, anxiety, novelty seeking, and wildness. In the 1970s, behavioral geneticists began utilizing recombinant inbred (RI) strains for behavioral genetic analysis [1]. RI strains made it feasible to perform quantitative trait locus (QTL) mapping, a powerful phenotype-driven method for the identification of genetic variants that significantly affect the variation of behavioral traits. One of the most widely used RI panels is the BXD set, derived from two founder strains, and this has been widely used to study the genetic basis of complex neurobehavioral traits [2,3]. However, the QTL mapping precision in studies using conventional RI lines is limited because they involve only two founder genomes, not enough to support variation among all traits, and the numbers of available strains generally did not allow high precision mapping of genes [4,5].

The CC overcomes these limitations. First proposed in 2004 [6], the CC is a powerful mouse RI panel specially designed for complex trait analysis. The eight-way “funnel” breeding scheme utilizes five classical strains (A/J, C57BL/6J, 129S1/SvlmJ, NOD/ShiLtJ, NZO/HILtJ) and three wild-derived strains (WSB/EiJ, CAST/EiJ, PWK/PhJ), to guarantee an abundance of genetic diversity with novel combinations of alleles to drive phenotypic diversity in almost any trait of interest [7]. The eight founder strains represent three subspecies of *Mus musculus*, capturing approximately 90% of the common genetic variation present in the *Mus musculus* species [8]. The CC population reaches a level of common genetic diversity and phenotypic diversity on par with the diversity found in the human population [9] and can allow identification of key mediator genes overlooked in much larger human studies [10,11]. The CC resource has low rates of long-range linkage disequilibrium across the genome and high rates of genetic recombination, allowing for accurate and precise localization of genetic loci; its high allelic diversity greatly increases the range of observable behavioral variation, improving the opportunities of detecting variation among traits and genetic mapping precision [4]. In the past decade, the CC resource has been used to study a wide range of complex traits, such as myocardial infarction [12], neurology disorders [13], tumors [14,15], diabetes [11,16], infectious diseases [17,18,19,20] and immune system traits [10,21]. 

Several behavioral studies using CC mice have been reported. Philip et al. utilized “pre-CC lines” (i.e., incipient CC lines which were not yet fully inbred) in a battery of behavioral tests [22]. Since then, the CC mice have been used to discover the relationship between rotarod performance and body weight, identifying 14 QTLs associated with body weight and 45 QTLs influencing rotarod performance [23]; investigating the quantitative genetic architecture of mouse behavior phenotypes used in neurodevelopmental disorder animal models; finding that digging, locomotor activity, and stereotyped exploratory patterns were mapped to QTLs harboring genes associated with corresponding phenotypes in human populations [24]; and measuring anxiety-like behavior by light/dark box assay, identifying candidate 141 genes significantly associated with anxiety [25].

In this study, we utilized 18 CC lines to perform a series of behavioral tests. This is the first study in which the CC mice have been characterized in the novel object recognition test, MWM task and fear conditioning test which assess anxiety level, learning and memory, respectively. The novel object recognition test, which draws on the natural tendency of rodents to investigate novel stimuli, evaluates non-spatial learning and memory of object identity. The MWM task is a swimming navigation test that assesses rodents’ spatial learning and memory. The fear-conditioning test is widely used to investigate mammalian fear learning and memory, as well as of pathological fear. We also applied the open field test to evaluate locomotor activity and anxiety state.

The complicated interactions among genetic background, environment, diet, nutritional supplements, drugs and lifestyle factors make research on learning and memory particularly difficult in human populations [26,27,28,29,30]. Applying the CC population to these learning and memory-related behavioral tests not only can overcome this problem by tightly controlling environmental and other factors to make a much more accurate analysis of the role of genetic background on learning and memory, but also can unravel genetic architecture of learning and memory considering the different aspects of these behaviors (i.e., non-spatial learning and memory; spatial learning and memory; fear learning and memory). 

## 2. Results

### 2.1. CC Behavioral Study Workflow

The workflow we utilized is illustrated in Figure 1. The first stage was to perform four behavioral assays (the open field test, novel object recognition test, MWM task and fear conditioning test) on 18 CC strains and four of the founder strains (A/J, C57BL/6J, 129S1/SvImJ and NOD/LtJ; these are hereafter referred to as B6, 129S1, and NOD, respectively). The second stage analyzed the resulting data by the following four steps: (i) visualization and statistical analyses of the data of each trait; (ii) mapping and generation of LOD-score plots for QTLs of each trait, assigning status to mapped QTLs depending on their genome-wide significant *p*-values: significant QTL(s) (*p* < 0.05), or “approaching significant” QTL(s) (*p*-value < 0.1), or suggestive QTL(s) (*p*-value < 0.63); (iii) for each chromosome with calculating coefficient plots showing the eight founder strains’ effects, generate box plots showing the association of founder haplotypes at peak SNPs and display founder haplotypes across the mapped loci; (iv) investigate sequence variation in candidate genes. All the findings of these genetic analyses were collected in the tables presented here.

### 2.2. Open Field Test Displays a High Phenotypic Diversity across the CC Strains 

Open field test performance varied widely across the 18 CC strains, indicating the influence of genetic diversity. Each of the parameters recorded during the test (time in the center of the open field in 5 min, time in the periphery of the open field in 5 min, time spent immobile in 5 min, distance traveled in the center of the open field in 5 min, distance traveled in the periphery of the open field in 5 min, total distance traveled in 5 min, showed statistically significant differences (*p* < 0.001) among the strains (Figure 2A–F). The strains LAM and A/J had extremely low levels of time and distance traveled in the center of the open field in 5 min, but extremely high levels of time and distance traveled in the periphery of the open field, a result strongly suggesting high anxiety levels (Figure 2A,B,D,E). In contrast, the 129S1 strain displayed high levels of time in the center of the open field in 5 min, but low levels of time and distance traveled in the periphery of the open field in 5 min, suggesting low anxiety levels (Figure 2A,D,E). A/J, LAM and 129S1 showed high levels of time spent immobile over 5 min, indicating passivity (Figure 2C).

### 2.3. Extensive Variation of Novel Object Recognition Test Traits among the CC Population

Four strains (LAM, A/J, 129S1 and LUV) showed very poor exploratory performances, either exploring only one or zero objects during the training phase, or exploring zero objects during the testing phase. Thus, these four strains were excluded from the final data analyses.

During the training phase of the novel object recognition test, CC strains displayed statistically significant differences in the time spent exploring the objects (*p* < 0.001) (Figure 3A) and a statistically significant difference (*p* < 0.001) between CC lines in the total time spent exploring two objects was also found (Figure 3C).

In the testing phase, a significant effect of strain was also observed on the time exploring the objects (*p* < 0.001), but there was no statistical difference (*p* = 0.35) across the CC lines in the time spent exploring the novel object and the familiar object (Figure 3B). Strains CIS, FEW, POH, DAVIS, and NOD spent longer times exploring the novel object than the familiar one, but the differences were not statistically significant (*p* > 0.05) (Figure 2B). There was a statistically significant difference between CC lines in the total time spent exploring two objects (*p* < 0.001) (Figure 3D). However, there were no statistical differences in the following three parameters: recognition index (*p* = 0.25) (Figure 3E) and discrimination index (*p* = 0.25) (Figure 3F). 

### 2.4. MWM Task Traits Are Widely Diverse in the CC Population

Strains A/J, 129S1, DAVIS, BEM, NOD and FIM were excluded from the final data analyses, because more than half the mice of each strain still could not learn to find the platform by the last day of the acquisition training. We observed that A/J mice floated quite frequently, swam slowly and pawed at the wall of the water tank quite a lot. 129S1 mice also had the problem of floating and sometimes swam slowly. All through the test, DAVIS and FIM mice demonstrated excessive thigmotaxis, i.e., the tendency to cling or follow the wall around the outer perimeter of the tank [31]. BEM mice swam slowly sometimes while NOD mice showed erratic performance.

The escape latency of SAT on the first day of the acquisition training was significantly (*p* < 0.05) different from the remaining 11 CC lines, thus it was also excluded from the final data analysis [31].

During acquisition training, two-way repeated measures ANOVA detected a significant effect of strain on escape latency to find the platform (*p* < 0.001) and thigmotaxis (*p* < 0.001), and also a significant effect of day on escape latency (*p* < 0.001) (Figure 4A) and thigmotaxis (*p* < 0.001) (Figure 4B). In addition, strain-by-day interaction effects (*p* < 0.001) were detected for escape latency to find the platform (Figure 4A) and for thigmotaxis (Figure 4B). All strains gradually decreased the latency to find the platform over the six training days, indicating they learned the task. BOM and POH learned very fast, and had much better performance than the other strains, while NUK, CIS and FEW learned slowest (Figure 4A). NUK showed statistically significantly higher thigmotaxis than other strains (Figure 4B). We also calculated the mean escape latency and thigmotaxis across the six training days, observing a significant effect of strain (*p* < 0.001) (Figure 4C,D). The result was consistent with the result shown in Figure 3A,B, that POH and BOM displayed shorter mean escape latency than other CC lines whereas NUK performed the longest mean escape latency (Figure 3C), and also showed the highest mean thigmotaxis (Figure 4D).

During the probe trial phase on Day 7, six parameters [thigmotaxis (Figure 4E), time spent in target quadrant (Figure 4F), frequency of crossing the platform area (Figure 4G), cumulative distance from the platform area (Figure 4H), time spent floating (%) (Figure 4I) and swim speed (Figure 4J)] were measured and analyzed with one-way ANOVA, detecting statistically significant differences across the CC lines (*p* < 0.001). NUK still displayed the highest thigmotaxis in the probe trial phase (Figure 4E). FEW and CIS presented shorter times in the target quadrant and the least frequency of crossing the platform area among the CC lines, while LAM, POH and BOM performed quite well with longer times in the target quadrant and higher frequency of crossing the platform area than others (Figure 4F,G). LAM displayed the shortest cumulative distance from the platform area (Figure 4H), a trait that has been suggested to be a good parameter for measuring spatial learning ability [31], and reflecting a more direct swimming trajectory of trying to find the platform. GIG, BOM and LAM showed a much shorter duration of time spent floating than other CC lines (Figure 4I). CIS and LAM had slower swim speed than others whereas BOM swam fastest of all CC lines (Figure 4J).

Factor analysis [32,33,34,35] of probe trial data from 94 subjects was taken to reduce the measurement of spatial memory confounded by the noncognitive factors. Three statistical factors were extracted, which could explain approximately 83.75% of the total variance in the probe trial performance (Figure 4K). Factor 1 (“thigmotaxis”), counted for 20.57% of the variability, had very strong positive factor loading (=0.931) for the duration of time spent within 10 cm of the perimeter of the tank. Factor 2 (“activity”), explaining 23.95% of the variability, had strong negative factor loadings (=−0.818) for the duration of floating, and also had strong positive factor loadings (=0.875) for the swim speed. Factor3 (“memory”), counted for 39.23% of the variability, had one very strong positive factor loading (=0.948) for the time spent in the target quadrant, another strong positive factor loading (=0.819) for the frequency of crossing the platform area and one strong negative factor loading (=−0.858) for the cumulative distance from the platform area. Thus 44.52% of the behavioral variability in the probe trial phase was accounted for by noncognitive factors, having no direct relationship with spatial memory.

A factor score for each subject was calculated, representing each subject’s normalized position in a particular factor. We detected statistically significant differences (*p* < 0.001) on scores of Factor 1 (Figure 4L), Factor 2 (Figure 4M) and Factor 3 (Figure 4N) across the CC lines using one-way ANOVA. LAM and NUK had the lowest and highest scores for “thigmotaxis”, respectively. CIS displayed the lowest score on “activity”, whereas BOM received the highest. FEW and CIS got lower scores for “memory” than any other CC lines, with POH, LAM and BOM showing higher scores than others.

On the cued learning phase, all the mice could finally find the platform, which suggested they had no severe vision impairment during the testing period.

### 2.5. Fear Conditioning Performance Covers a Wide Range in the CC Population

We observed substantial continuous variation of 27 parameters measured in the fear conditioning test, with significant differences (*p* < 0.001) across the 18 CC lines.

The following 13 parameters were used as indices of fear conditioning acquisition: the percentage of time spent freezing during the first, second, third, fourth and fifth tone shock interval, the percentage of time spent freezing during the total five tone shock intervals, the percentage of time spent freezing during the first, second, third, fourth and fifth post-shock interval, the percentage of time spent freezing during the total five post-shock intervals, and the percentage of time spent freezing during 181–600 s on Day 2 for fear conditioning training (Figure 5A). BOM and FEW had lower fear conditioning acquisition levels than other CC lines, while 129S1 and YID showed higher levels. 

The percentage of time spent freezing during 0–330 s on Day 3 for contextual fear conditioning was used as an index of contextual fear memory (Figure 5B). 129S1, SAT and POH displayed more excellent contextual fear memory than other CC lines, while BOM and NOD had poorer contextual fear memory than others.

13 parameters (namely the percentage of time spent freezing during the first, second, third, fourth and fifth tone shock interval, the percentage of time spent freezing during the total five tone shock intervals, the percentage of time spent freezing during the first, second, third, fourth and fifth post-shock interval, the percentage of time spent freezing during the total five post-shock intervals, and the percentage of time spent freezing during 181–600 s) on Day 4 for cued fear conditioning were used as indices of cued fear memory (Figure 5C). PIPING, GIG, LUV and BEM showed lower cued fear memory levels than other CC lines. In contrast, 129S1, SAT and LAM showed much higher levels of cued fear memory.

### 2.6. Genome-Wide Association Analysis for Behavioral Phenotypes Identified Significant QTLs

We performed genome-wide interval mapping on 52 behavioral traits using the GeneMiner platform [36,37]. Collectively, we identified 3 significant (*p* < 0.05), 3 approaching significant and 81 suggestive QTLs (Table 1 and Appendix A). The summaries of each QTL are shown in Appendix A. 

The first significant QTL was associated with the frequency of crossing the platform area on the probe trial phase of MWM, which reflects the spatial memory level (Figure 6A; Table 1). This QTL, hereafter referred to as Spatial memory locus-2 (*Sml2*), falls into the region 128.5–135.0 Mbp on chromosome (Chr) 4, producing a maximum LOD of 12.9. Founder haplotype analysis revealed that the CAST/EiJ (CAST) founder strain had predominant influences on this trait, negatively influencing the frequency of crossing the platform area on the probe trial phase (Figure 6B). The imputed haplotypes for 18 CC lines showed that the strains that exhibited lower frequency of crossing the platform area on the probe trial phase (FEW, CIS) inherited CAST alleles at this locus (Figure 6C). A search of the Sanger database [38] identified 57 candidate genes possessing CAST-specific protein-changing single-nucleotide polymorphisms (SNPs) or insertions-deletions (Indels), including 53 coding genes, 3 noncoding genes and 1 unclassified gene (Appendix A). Of these, the most plausible candidate genes (*Lck*, *Sdc3*, *Ox1r*, *Ptafr*, *Srsf4*, *Stx12*, *Phactr4*) carrying missense variants specific to CAST will be reviewed in the Discussion section. 

The second significant QTL was for swim speed on the probe trial phase of the MWM, which reflects locomotor activity. This locus (hereafter referred to as Locomotor activity locus-3, *Lal3*) mapped to Chr15, spanning the region 88.05–99.35 Mbp with a maximum LOD of 12.1 (Figure 7A; Table 1). Examination of the founder haplotype coefficients in this interval indicated that the causal variants were derived from the 129S1 and CAST founder strains (Figure 7B). The CIS and LAM strains had 129S1 and CAST haplotypes in this interval, respectively, and displayed the slowest swim speed on the probe trial phase (Figure 7C). A search of the Sanger database identified 51 candidate genes possessing 129S1 and/or CAST-unique protein-changing SNPs/Indels, including 50 protein coding genes and 1 noncoding gene (Appendix A). Of these, ten genes (*Cntn1*, *Prickle1*, *Kif21a*, *Lrrk2*, *Kmt2b*, *Hdac7*, *Senp1*, *Plxnb2*, *Sbf1*, *Nell2*) with missense variants would be the best candidates to explain this QTL.

The third significant QTL (hereafter referred to as *Cued fear memory locus-1*, *Cfml1*) encompassing a region (Chr4: 127.45–132.63 Mbp) with a maximum LOD of 9.0 was associated with the percent of time spent freezing during the first post-shock interval on Day 4 of the fear conditioning test for cued fear conditioning, which reflects the cued fear memory level (Figure 8A; Table 1). Both the NOD and 129S1 alleles were associated with increased percent of time spent freezing during the first post-shock interval in the cued fear conditioning test, while PWK/PhJ [hereafter referred to as PWK) allele was related to decreased time (Figure 8B). Five CC strains (SAT, POH, YID, NOD) with the NOD or 129S1 alleles in this QTL interval presented lower percent of time spent freezing during the first post-shock interval in the cued fear conditioning test, while those (BOM, DAVIS) with the PWK alleles exhibited higher percentages (Figure 8C). A search of the Sanger database identified 38 candidate genes possessing 129S1 and/or CAST-unique protein-changing SNPs/Indels and PWK-specific protein-changing SNPs/Indels, including 36 coding and 2 noncoding genes (Appendix A). Of these, we found 16 genes that were candidates for regulating both spatial memory level and condition fear memory level traits (i.e., the *Sml2* and *Cfml1* QTLs). Among the 38 candidate genes, two genes (*Ak2*, *Hpca*) carrying splice region variants and four genes (*Phactr4*, *Sdc3*, *Azin2*, *Hdac1*) carrying missense variants seemed to be the most likely candidates to explain this *Cfml1* QTL. 

The first approaching significance QTL (hereafter referred to as *Locomotor activity locus-4*, *Lal4*), falling into the region 149.98–152.61 Mbp on Chr4, affected the time spent immobile in an open field during a 5 min test period in the open field test, which reflects the locomotor activity level (Appendix A; Table 1). The 129S1-derived variant was associated with increased time spent immobile in an open field during a 5 min test period, whereas the NOD/LtJ-derived variant was associated with decreased time (Appendix A). The NOD/LtJ founder strain and FEW strain with the NOD/LtJ allele at this locus exhibited shorter time spent immobile in an open field during a 5 min test period, while the 129S1 founder strain and LAM strain with 129S1 alleles at this locus exhibited longer times (Appendix A). Three candidate genes (*Tnfrsf9*, *Per3*, *Camta1*) were identified at this locus (Appendix A); all are protein-coding genes carrying SNPs specific to the 129S1 strain.

For the total distance traveled in the periphery of an open field during a 5 min period in the open field test (reflecting the anxiety level), we identified the second approaching significance QTL (hereafter referred to as *Anxiety level locus-4*, *All4*) on Chr7, spanning the region 149.55–152.51 Mbp (Appendix A; Table 1). Appendix A illustrated that the A/J and 129S alleles were associated with decreased distance traveled in the periphery of an open field during a 5 min period, while PWK alelles were associated with increased distance. Four strains (129, LAM, A/J, BEM) inherited the A/J or 129 alleles at this locus and exhibited shorter distance traveled in the periphery of an open field during a 5 min period, while BOM and YID strains inheriting the PWK alleles at this locus exhibited longer distance (Appendix A). We identified 16 candidate genes carrying protein-changing SNPs/Indels specific to the A/J and/or 129S1 background and protein-changing SNPs/Indels unique to the PWK background, including 15 coding genes and 1 noncoding gene (Appendix A). Of these, four genes (*Shank2*, *Ctsd*, *Igf2*, *Th*) carrying missense variants have been reported to be associated with anxiety. 

Three cued fear memory indexes, the percentage of time spent freezing during the third post-shock interval (Appendix A), the fifth post-shock interval (Appendix A) and total five post-shock intervals on Day 4 of the fear conditioning test (Appendix A), were driven by the same QTL—the third approaching significance QTL (hereafter referred to as *Cued fear memory locus-2*, *Cfml2*) on Chr2 in the interval of 156.60–170.90 Mbp with a maximum LOD of 8.8 (Appendix A; Table 1). Both the 129S1 and A/J alleles had clear negative contributions on these three traits (Appendix A). Two strains (129S1 and SAT) inherited 129S1 alleles and another two strains (LAM and A/J) inherited A/J alleles at this locus had higher percent of time spent freezing during the third post-shock interval, the fifth post-shock interval and total five post-shock intervals in the cued fear conditioning test, respectively (Appendix A). We identified 24 candidate genes possessing 129S1 and/or A/J-unique protein-changing SNPs/Indels, including 22 protein coding genes and 2 noncoding gene (Appendix A). Of these, six genes (*Src3*, *Neurl2*, *Elmo2*, *Sulf2*, *Pigt*, *Ctnnbl1*) carrying missense variants and two genes (*Slc12a5*, *Cd40*) carrying splice region variants seemed to be the best candidates. 

Additionally, 81 QTLs were identified in this study with “suggestive” statistical support (Appendix A). Characterization of additional CC strains could be performed to confirm or reject these QTLs.

## 3. Discussion

In the present study, four behavioral tests were conducted using CC mice. This is the first time that CC strains had been characterized in three of these tests, namely novel object recognition tests, Morris water maze tasks, and fear conditioning tests. Reflecting their genetic diversity, there was a wide range of phenotypic diversity amongst the CC mice in performing these tests. Reflecting the great genetic variability captured by the CC, we found substantial continuous behavioral variations across the CC strains on locomotor activity, anxiety level, non-spatial/spatial/fear learning, and spatial/fear memory. We discuss below the responses of the CC strains in these tests, as well as candidate genes that may mediate the relevant traits.

Mice of the A/J strain showed poor swimming ability, long times spent floating and low levels of exploration, and this was compatible with their performances in the open field test in which they spent long times immobile, traveled short total distances in the open field in the 5 min test time, and also compatible with their poor exploratory performances in the novel object recognition test. A possible reason might be that A/J mice have mutations in the gene encoding the skeletal muscle protein dysferlin, producing a group of muscle degenerative disorders [39,40,41,42,43,44]. Similar deficiencies of dysferlin were reported to be correlated with muscular dystrophy in humans, leading to clinically distinct muscle diseases, including limb girdle muscular dystrophy type 2B (LGMD 2B) with predominantly proximal weakness, Miyoshi myopathy (MM) with calf muscle weakness and atrophy, and distal myopathy with anterior tibial onset (DMAT) with tibialis muscle atrophy [45,46,47,48]. We could observe that A/J mice persistently hugged the tank wall in the MWM task, indicating high anxiety levels, compatible with their performances in the open field test that they spent short time in the center of the open field and long time in the periphery of the open field during the 5 min test time.

Most 129S1 mice in our study showed a relatively high frequency of floating and slow swim speed, and twelve 129S1 mice of a total of twenty tested failed in learning to find the platform on the last day of the acquisition training in the MWM task. Surprisingly, several 129S1 mice did a good job in the MWM task. About 70% mice of the 129/J substrain were reported to lack a corpus callosum, which might be related to retarded formation of the hippocampal commissure in embryos [49]. We dissected the brains of twenty 129S1 mice after the behavioral tests and found that each mouse had a corpus callosum. Thus, the variable performances of the 129S1 mice showed in the MWM task were not due to lacking a corpus callosum. In the open field test, 129S1 mice spent a remarkably long time in the center of the open field and short times in the periphery of the open field, suggesting very low levels of anxiety. They also demonstrated low levels of exploration in the open field and spent a remarkably long time immobile, compatible with their really poor performance in the novel object recognition test. They were excluded from the final data analyses of the novel object recognition test as well. Conversely, 129S1 mice showed really good performance in the fear conditioning test, with high fear conditioning acquisition index, high contextual fear memory index and cued fear memory index.

We observed that most albino strains in our study (A/J, NOD, CIS, NUK and FEM) performed relatively worse than other CC lines in the MWM task where they had very poor performance on the acquisition training and probe trail phases. Many albino strains have been reported to be impaired visually [50,51,52,53]. However, this trait was unlikely to have contributed to their poor performance since genetic mapping did not show significant linkage to the albino locus on chromosome 7. In the cued learning phase of the MWM task, all the mice were eventually able to find the platform, demonstrating any visual impairment did not affect them in the testing period. Furthermore, most of these strains (except A/J) had equally bad performance in fear conditioning acquisition and cued fear conditioning, which were not dependent on vision.

Factor analysis of the MWM data from all 94 subjects extracted three statistical factors which could explain 83.75% of the total variance, and is consistent with previous studies [32,33,34,35]. Our the factor analysis was conducted only using probe trial data while others performed used both spatial acquisition data and probe trial data, so did not separate the spatial learning and memory processes but treated the spatial acquisition and probe trial phases as a whole when performing the factor analyses. We measure the level of memory, so we only used the probe trial data for the factor analysis.

Mice displaying thigmotaxis are not problem-solving [31]. Sometimes, a “non-spatial” pre-training procedure could reduce excessive thigmotaxis [31]. Lipp and Wolfer did a similar factor analysis using MWM data from 1538 mice and found that 48% of the variance was explained by Factor “thigmotaxis” [32]. We observed substantial continuous variation of thigmotaxis in CC lines during our testing. We detected four suggestive QTLs for thigmotaxis, one using the mean thigmotaxis during spatial acquisition phase data, while the other three used the scores for Factor “thigmotaxis” data. 

In our study, QTLs mediating the behavioral traits exhibited in these tests were identified and candidate genes within the QTLs were detected. The *Sml2* QTL regulated spatial memory. Of the 57 candidate genes within *Sml2*, seven were considered as the best candidates either because they have been reported to be associated with spatial memory, or could be related to other genes for this trait according to previous studies. These seven genes are discussed as follows. *Lck* was reported to regulate long-term synaptic strengthening, hippocampus-dependent spatial learning and memory in mice [54], and is proposed to be a risk gene for Alzheimer’s disease (AD) in humans [55]. SDC3, a transmembrane heparan sulfate proteoglycan, was suggested to act as an important modulator of synaptic plasticity that influences mouse hippocampus-dependent spatial memory [56], and also could serve as an emerging molecular target in AD in mice [57]. OX1R, the receptor of orexin A, was shown to modulate mouse hippocampal CA1 synaptic plasticity [58] and play an important role in mouse cue-dependent fear memory acquisition and consolidation [59]. Furthermore, OX1R has been proposed to play a significant role in spatial learning and memory in rats [60,61]. *PTAFR* was suggested to be related to AD in humans via the abnormality of signal transduction [62,63]. SRSF4, serine/arginine-rich splicing factor 4, interacting with heterogeneous nuclear ribonucleoproteins hnRNPG and hnRNPE2, regulates the 5’ splice site of tau exon 10; splicing misregulation could lead to frontotemporal dementia (FTD) in humans [64]. STX12, syntaxin 12, might also cause FTD in humans; its interaction with the v-SNARE protein VAMP3 is enhanced by TANK-binding kinase 1 (TBK1) haploinsufficiency [65]. *Phactr4*, phosphatase and actin regulator 4, forms functional associations with intermediate filament networks in mouse adult neural stem cells (NSCs) and their progenitor astrocytes [66], as well as playing an important role in the reactive gliosis following mouse brain injury [67]. *PHACTR4* was also reported to be implicated in AD in humans [68].

The *Cfml1* and *Cfml2* QTLs both regulated cued fear memory. Fourteen of the 62 candidate genes in these two regions were considered as the top candidates; two of these have been reported to play unique roles in contextual fear learning or memory, while the others have a plausible role in cued fear memory. A previous study has suggested that HDAC1 (a member of class-I HDAC proteins) in the adult mouse hippocampus plays a specific role in the extinction of contextual fear memories, regulating extinction learning via a mechanism involving H3K9 deacetylation and subsequent trimethylation of target genes under physiological conditions [69]. SRC3, a member of the steroid receptor coactivator (SRC) family, has been demonstrated to regulate synaptic plasticity and contextual fear learning via acetylating calmodulin in the mouse brain [70]. It is controversial whether *CTNNBL1* is a human episodic memory-related gene [71], but nevertheless, our results suggest that *Ctnnbl1* could mediate cued fear memory in mice. Hence, more studies are required to test the potential role of the *CTNNBL1* gene in human memory. Deep proteome profiling of undepleted human serum suggested that mitochondrial protein AK2 may serve as a novel AD candidate biomarker [72]. Hippocalcin (HPCA), a member of the neuronal calcium sensor protein family, functions as a calcium sensor in hippocampal long-term depression (LTD) [73]. A previous study showed that Hpca-deficient mice displayed impaired spatial and associative memory in the MWM task and discrimination learning task, respectively, while no abnormalities were observed in fear learning via the passive avoidance learning test [74]. Hence, further investigation is required for demonstrating the role of HPCA in cued fear memory. Sustained AZIN2 (antizyme inhibitor 2) overexpression in the mouse brain has been demonstrated to increase acetylpolyamines and precipitate tau neuropathology, which induces cognitive and affective behavioral impairments [75]. The *Neurl2* gene was suggested to be crucially involved in mouse hippocampus-dependent spatial memory and synaptic plasticity [76]. The mouse *Elmo2* gene, was identified to induce neurite outgrowth [77], which is closely involved with learning and memory [78]. sulfatase 2 (SULF2) expression was found to be reduced in AD patients’ specific regions of the brain, which suggested that SULF2 might be associated with the pathogenesis of AD [79]. Homozygous mutations in the human PIGT gene leading to glycosylphosphatidylinositol (GPI) anchor deficiency was reported to cause a novel intellectual disability syndrome [80]. Transmembrane protein CD40, a member of tumor necrosis factor receptor superfamily, initiates the AD pathogenesis and exacerbates the AD progression [81]. K^+^-Cl^−^ cotransporter 2 (KCC2) encoded by the *Slc12a5* gene, is a neuron-specific cotransporter. Previous study showed that altering KCC2 function caused long-term abnormalities in mouse spatial memory retention [82]. Reduced KCC2 expression interferes with synapse specificity of LTP in aged mouse brains, indicating the potential mechanisms underlying the decreased memory in aged mice [83]. Overexpressed KCC2 improved mouse motor learning rates [84]. Notably, *Sdc3* and *Phactr4*, which have been described as the candidates regulating spatial memory above, were also the candidates regulating cued fear memory. Thus, these two genes were suggested to be candidates related with memory by two independent behavioral tests, which indicated great potential for regulating memory. Of note, a report in 2002 showed that *Sdc3*-knockout mice displayed impaired contextual fear memory but had similar cued fear memory as their wild-type littermates [56]. Thus, further investigations are still needed to confirm whether the *Sdc3* gene plays a role in cued fear memory.

The *Lal3* and *Lal4* QTLs both regulated locomotor activity. Of the 54 candidate genes in these two regions, 13 seemed to be the most plausible according to previous studies. In humans, the *CNTN1* gene, encoding contactin-1, a neural adhesion and neuromuscular junction (NMJ) protein, whose mutations can impair communication or adhesion between nerve and muscle leading to severe myopathic phenotype, as severe congenital myopathy with fetal akinesia (FA) and nonspecific myopathic features on muscle biopsy [85]. *Cntn1* null mice exhibited severe ataxia, progressive muscle weakness and growth retardation [86]. Contactin-1 was reported to play a critical role in the regulation of myelination, axogenesis and fasciculation [87]. In humans, individuals with homozygous mutations in the gene *PRICKLE1* typically present with progressive myoclonus epilepsy (PME) and motor impairment with ataxia [88]. *PRICKLE1* was also suggested to be implicated in FA via next-generation sequencing (NGS) techniques [89]. Similarly, bi-allelic loss-of-function *KIF21A* variants have been described to cause severe neurogenic FA [90]. Kinase activating missense mutations in *LRRK2* represent the largest known cause of Parkinson’s disease (PD) with corticobasal degeneration and associated motor neuron disease in some patients [91,92]. HDAC7, a member of class-II histone deacetylase (HDAC) proteins, plays a dynamic role in MEF2-mediated skeletal mouse muscle differentiation [93], and the mRNA level of Hdac7 decreases in mouse atrophying skeletal muscle [94]. Mutations in the histone lysine methyltransferase gene KMT2B has been found leading to early onset generalized dystonia in humans [95]. The protein SENP1 was proposed to be one novel isoform of the human tendon biomarker tenomodulin (TEMD) [96]. A previous study of *Tnmd* knockout mice demonstrated that TEMD deficiency causes hampered running performance, and TEMD is essential to tendon endurance running [97]. The mouse *Plxnb2* gene was reported to control cerebellar granule cells’ development [98], and the human *PLXNB2* gene was suggested to be relevant to amyotrophic lateral sclerosis (ALS) pathogenesis [99]. Mutations in the human *SBF1* gene were identified to cause motor and sensory neuropathies [100,101]. The secreted glycoprotein, neural epidermal growth factor-like like 2 (NELL2), was reported to promote motor neuron differentiation in the chick dorsal root ganglion (DRG) [102]. Güler et al. reported a novel deletion in the neighboring genes *DJ-1* and *TNFRSF9* in three siblings of a Turkish family, whose ages of early onset PD onset were remarkably earlier than patients only possessing the *DJ-1* deletion [103]. This report suggested that *TNFRSF9* might be a possible disease modifier, possessing the potential to modify the phenotype of early onset PD [103]. A variable number tandem-repeat (VNTR) polymorphism in the *PER3* gene related with immune response [104], circadian rhythm phenotypes and homeostatic regulation of sleep [105], might accelerate the disease course in Multiple Sclerosis (MS) patients by disrupting circadian cycle of sleep [106]. Mice lacking the CAMTA1 (calmodulin-binding transcription activator 1) transcription factor had shown severe ataxia and Purkinje cell degeneration [107]. Mutations and intragenic rearrangements in the *CAMTA1* gene result in non-progressive cerebellar ataxia with or without intellectual disability and attention deficit hyperactivity disorder (ADHD) in humans [108,109,110]. A myoclonic dystonia-predominant phenotype was also described which result from a novel *CAMTA1* sequence variant [111]. A variant in *CAMTA1* was also identified to be associated with ALS patients’ survival [112].

We detected 16 candidate genes in the *All4* QTL, which regulated anxiety level. Of these, the following four genes have been reported to be related with anxiety. Cerebellar Shank2 (an excitatory postsynaptic scaffolding protein) was proposed to regulate specific repetitive and anxiety-like behaviors [113]. Mice heterozygous for cathepsin D deficiency (CTSD) manifest reduced anxiety-like behavior [114]. Igf2-inducible knockout mice displayed increased anxiety [115]; *IGF2* was shown to be one of the anxiety-related differentially methylated genes in humans [116]. The activity of tyrosine hydroxylase (TH) regulates dopamine [117], whose levels and metabolites in the mouse brain are related with anxiety-like behavior [118].

A potentially interesting observation is that three QTL intervals, including two of the three significant QTLs, contain genes encoding proteins that bind zinc: *Zbtb8b*, *Zdhhc18*, *Zfp683* and *Zfp593* genes in the *Sml2* QTL associated with spatial memory, the *Zscan20*, *Zbtb8os*, *Zbtb8a*, *Zbtb8b*, *Zfp663*, *Zfp334* and *Zmynd8* genes in the *Cfml1* and *Cfml2* QTLs associated with cued fear memory. Zinc finger proteins might play a critical role in cognition.

Our results defining substantial continuous phenotypic variations across the CC panel demonstrate that genetic variation in mice does influence behavioral traits, including learning, memory, locomotor activity and anxiety levels. Consistent with the requirement of translational research that mouse models should accurately reflect human conditions, this marked phenotypic diversity demonstrates immense potential for the identification of novel genes and molecular mechanisms to understand these behavioral traits in humans. 

## 4. Materials and Methods

### 4.1. Mice

14 CC strains and 4 founder stains were obtained from the Institute of Experimental Animals of the Chinese Academy of Medical Science (Beijing, China). At least nine mice from each strain were tested for the behavioral measures. Only male mice were used in this study to avoid the influence of the 4–5 day estrous cycle of female mice during the course of behavioral experiments [119]. Mice were housed under specific pathogen-free conditions and were provided with standard rodent food and tap water ad libitum. All cages were kept under the same conditions, maintained under constant temperature (21–22 °C) and humidity (55 ± 5%) conditions, with a standard 12:12 light:dark cycle. All mouse breeding and experimental procedures were conducted in compliance with the Institutional Guidelines for the Care and Use of Laboratory Animals, Institute of Zoology (Beijing, China), and were approved by the Ethics Committee of the Laboratory of Animal Science of Peking Union Medical College, and (Approval NO. QC20011).

### 4.2. General Behavioral Testing Procedures

The sequence of the four behavioral experiments was arranged as follows: open field test, novel object recognition test, MWM task, fear conditioning. Mice were at 10 ± 1 weeks of age on the first day of testing. The body weight of the mice is between 16.7–40.0 g. Mice were habituated to the testing room for 1 h prior to testing. The same experimenter handled the mice for each test and the same individuals were in the room during all sessions of a particular test. Behavioral measures of open field test, novel object recognition test and MWM task were recorded and analyzed by Ethovision XT real-time video tracking system (Noldus Information Technology, Wageningen, The Netherlands). All activities of fear conditioning were scored and assessed by Video Freeze system (Med Associates, Inc., St. Albans, VT, USA).

#### 4.2.1. Open Field Test

Mice naturally tend to stay at the border area of an open field. Longer times spent in the center indicates lower anxiety states. The apparatus for the open field test is a behavioral box 50 cm in length, 50 cm wide, and 30 cm high, divided into three zones—center, 20 × 20 cm^2^; intermediate zone, (30 × 30–20 × 20) cm^2^ and periphery, (50 × 50–30 × 30) cm^2^. Each mouse was gently placed into the center of the arena and allowed to freely explore for 5 min. The apparatus was thoroughly cleaned between trials using 75% vol/vol ethanol to avoid olfactory cues. The following parameters were recorded: time in the center and periphery of the open field in 5 min, time spent immobile in 5 min, distance traveled in the center and periphery of the open field in 5 min, and total distance traveled in 5 min. 

#### 4.2.2. Novel Object Recognition Test

Mice with good memory of a familiar object will spontaneously prefer exploring a novel object. The apparatus is the same box used in the open field test. Three phases were included in the task: habituation phase, training phase, and testing phase. The habituation phase was performed on the first day; we saw the open field test as the habituation phase. The training and testing phases were performed on the second day, with an interval of one hour between them. Two identical objects (A1 and A2) were placed in the opposite corners of the arena (NW corner and SW corner) during the training phase. During the testing phase, almost half mice of each strain were put in the apparatus so that A1 was substituted by one novel object, while the other half were put in the apparatus with A2 being replaced by the same novel object. This measure could eliminate the influence of mice on location preference. Each mouse was gently placed into the center of the arena and allowed free exploration for 5 min on both training phase and testing phase. The box and objects were thoroughly cleaned between trials using 75% vol/vol ethanol to avoid olfactory cues. Exploration was defined as when the mice’s noses pointed towards the object and within 1.5 cm of the object, with active sniffing. Parameters were measured as follows: time spent exploring object A1, time spent exploring object A2, total time spent exploring object A1 and object A2 on training phase (TA1 + TA2); time spent exploring novel object, time spent exploring familiar object, total time spent exploring novel object and familiar object (TN + TF), time spent exploring novel object minus time spent exploring familiar object (TN − TF) on testing phase. Discrimination index (DI) and recognition index (RI) were calculated using the following equation:DI = (TN − TF)/(TN + TF), RI = TN/(TN + TF).

In addition, a mouse was excluded from the final data analyses if it did not explore both objects during the training phase, or if it did not explore either object during the testing phase.

#### 4.2.3. MWM Task

The open circular water tank is 120 cm in diameter, is filled with water (19–22 °C temperature) and opaque by white (when albino strains were in the test) or black (when non-albino strains were in the test), non-toxic paint. Two principal axes bisecting the tank perpendicular to one another were designated, creating an imaginary “+”. In this way, the tank was divided into four equal quadrants, which were named as NE, SE, SW and NW in compass locations. The platform (circular, 10 cm diameter) was placed in the middle of the NW quadrant. 

Three phases were arranged in the test, and the sequence was: spatial acquisition, probe trial and cued learning. 

18 learning trails were conducted over 6 days in the spatial acquisition phase, with 3 trials per day. After all mice had completed Trial 1, the next trial would begin. 10 min were taken in the interval between trials per day. The platform was submerged 1 cm below the water surface. Mice were released into the water at water-level (not dropped) with their heads facing the tank wall at three start positions of NE, SE, SW per day. The order of start positions was different each day. A maximal time of 60 s was allowed for a mouse to find the submerged platform, and if stayed on it for at least 5 s it was defined as “successful” in this task. After staying on the platform for 15 s, the mouse was gently picked up and removed to its cage, allowed to warm up and dry off using an electric warmer. If a mouse failed to find the platform within 60 s, it would be guided to the platform and stayed on the platform for 15 s. 

24 h after the last spatial acquisition day, mice received a probe trial. The platform was removed. Mice were released into the water at water-level with their heads facing the tank wall at the start position of 180° from the original platform position and allowed to swim freely in the tank for 60 s.

1 h after the probe trial, the cued learning phase performed. The platform was elevated 1 cm above the water surface, and a ‘flag’ was mounted that extends above the water surface by 10 cm. This cue would allow the mice a direct line-of-sight to the platform’s location. Mice were released into the water at water-level with their heads facing the platform at the start position of NW and allowed to swim freely in the tank for 60 s. A maximal time of 5 min was allowed for a mouse to find the platform, and needed to keeping staying on it for at least 5 s for task success.

Parameters were measured as follows: escape latency (s) to find the platform and the duration of time spent within 10 cm of the perimeter of the tank (‘thigmotaxis’: percent of total time, chance level = 30.56%) on each day during acquisition training; thigmotaxis, time spent in the target quadrant, frequency of crossing platform area (count/min), cumulative distance (m) from the platform area (the sum of the distances from the platform sampled once every 0.01 s throughout the trial), time spent floating (%), and swim speed during probe trial. 

In addition, if more than half of the mice of a strain still could not learn to locate the platform on the last day of the acquisition training, that strain would be excluded from the final data analyses. If the escape latency of one strain on the first day of the acquisition training was significantly different from other strains, that strain would be excluded from the final data analysis [31].

#### 4.2.4. Fear Conditioning Test

The fear conditioning test was performed using Med Associates Video Freeze fear conditioning equipment (Med Associates, Inc., St. Albans, VT, USA). The test consisted of contextual fear conditioning test and cued fearing conditioning test. 

Each mouse was placed in a conditioning chamber and for adaptation was allowed to explore the context freely for 10 min on Day 1. On Day 2, fear conditioning training consisted of a 180 s baseline period followed by five pairings of a tone (30 s, 5 kHz, 80 dB) that precedes and co-terminates with a mild foot shock (1 s, 0.5 mA), separated by 60 s intervals. On Day 3, the contextual fear conditioning test was conducted. Each mouse was placed into the same conditioning chamber for 330 s, with no auditory or electric current stimulation. On Day 4, the background and smell of the conditioning chamber were all changed, the cued fear conditioning test consisted of a 180 s baseline period followed by five pairings of a tone (30 s, 5 kHz, 80 dB). 

The apparatus was thoroughly cleaned with 70% *vol*/*vol* ethanol before the placement of each animal during the four days.

We measured fear conditioning by freezing behavior, a reliable measure of fear in rodents, defined as an innate defensive behavior defined as complete immobility with the exception of respiratory movements [120,121].

Parameters were measured as follows: the percentage of time spent freezing during the first, second, third, fourth and fifth tone shock interval, the percentage of time spent freezing during the total five tone shock intervals, the percentage of time spent freezing during the first, second, third, fourth and fifth post-shock interval (a 30 s interval following each foot shock was defined as the post-shock interval), the percentage of time spent freezing during the total five post-shock intervals, and the percentage of time spent freezing during 181–600 s on Day 2; the percentage of time spent freezing during 0–330 s on Day 3; the percentage of time spent freezing during the first, second, third, fourth and fifth tone shock interval, the percentage of time spent freezing during the total five tone shock intervals, the percentage of time spent freezing during the first, second, third, fourth and fifth post-shock interval, the percentage of time spent freezing during the total five post-shock intervals, and the percentage of time spent freezing during 181–600 s on Day 4.

### 4.3. Data Analysis

Statistical differences between CC lines were evaluated using two-way repeated measures (mixed model) ANOVA to compare the time exploring the two objects in the training phase and the testing phase in the novel object recognition test [122]. Paired *t*-tests were performed to compare the time spent exploring the novel object and the familiar one in the testing phase.

Escape latency and thigmotaxis data of learning acquisition phase in MWM for each subject were subjected to an 11 (strain) x 6 (day) two-way repeated measures ANOVA with Bonferroni’s post hoc test, with strain as a between-subjects factor and day as a repeated factor. 

Factor analysis using a principal components solution with orthogonal (varimax) rotation was used to reduce redundancy of behavioral parameters in probe trial phase of MWM, in order to separate the of cognitive and non-cognitive factors statistically [32,33,34,35]. Boxplot analysis was employed as a method for outlier detection. Then, two subjects were deleted from the factor analysis because of containing extreme outliers. 94 subjects were counted into the analysis totally at last. Factor analysis is a way that could realize the recognition the clusters of strongly inter-correlated variables, which is called factors. Loadings (−1.0~1.0) showed in the table are the correlation coefficients of each variable with the factors, which could suggest the degree that one variable is correlated with the factors. The factor scores of each subject could also be calculated. One-way ANOVA test was used to detect the effect of strain on each factor [35].

Except for the above, all the rest of the behavioral parameters in the study (open field test, novel object recognition test, MWM, fear conditioning) were analyzed using one-way ANOVA test. Boxplot analysis was also applied to delete the outliers. Additionally, in every test, the outliers in each parameter were deleted separately, irrespective of the relationship between the parameters.

All analyses were performed using IBM-SPSS Statistics software (version 25; IBM, Armonk, NY, USA). Differences were considered statistically significant at the *p* < 0.05, *p* < 0.01 and *p* < 0.001 levels. Graphs were generated using GraphPad Prism 8 (GraphPad Software, La Jolla, CA, USA). Data are expressed as means ± S.E.M.

### 4.4. QTL Mapping

The GeneMiner online platform was used for QTL mapping. On this platform, linkage analysis could be performed using inferred founder haplotypes to identify QTLs. A logistic regression model was used to calculate the maximum-likelihood estimate (derived logarithm of odds [LOD] score) for each genomic position. Genome-wide permutations with *p* value thresholds of *p* < 0.05, *p* < 0.10 and *p* < 0.63 were used to define significant QTLs, approaching significant QTLs and suggestive QTLs, respectively [37]. A LOD 2—drop interval from the peak position was defined as the 99% confidence interval of each QTL. Candidate genes containing variants which contribute to protein changes (altering gene product structure or function, such as missense, frameshift and splice region variants) within each QTL region were identified using genome sequences from the Wellcome Trust Sanger Institute Mouse Genome Project [38].

## Figures and Tables

**Figure 1 ijms-24-00682-f001:**
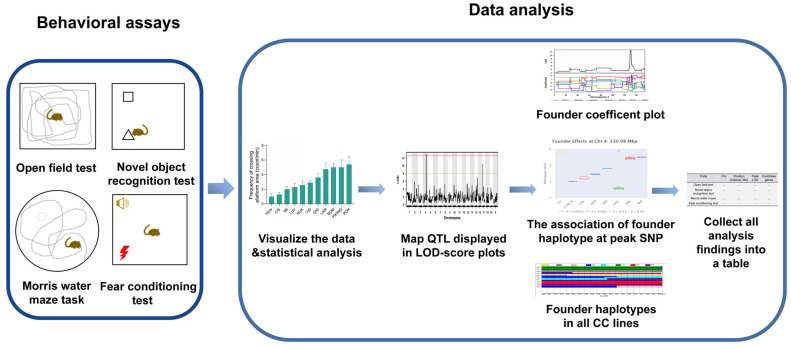
Illustration of the CC behavioral study workflow.

**Figure 2 ijms-24-00682-f002:**
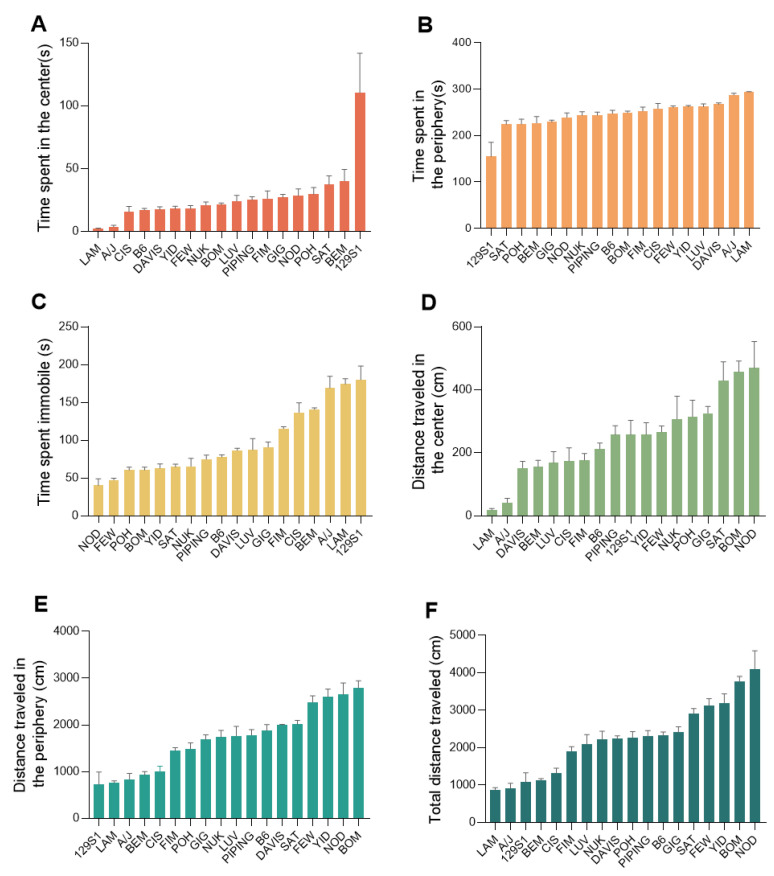
Phenotypic diversity in open field traits among the CC population. (**A**) Time spent in the center of an open field during a 5 min test period (s). (**B**) Time spent in the periphery of an open field during a 5 min test period (s). (**C**) Time spent immobile of an open field during a 5 min test period (s). (**D**) Distance traveled in the center of an open field during 5 min test period (s). (**E**) Distance traveled in the periphery of an open field during 5 min test period (s). (**F**) Total distance traveled in an open field during a 5 min test period (cm). Plots are expressed as mean ± SEM, with CC lines ordered along the *x*-axis by mean per phenotype. ((**A**–**F**) one-way ANOVA *p* < 0.001).

**Figure 3 ijms-24-00682-f003:**
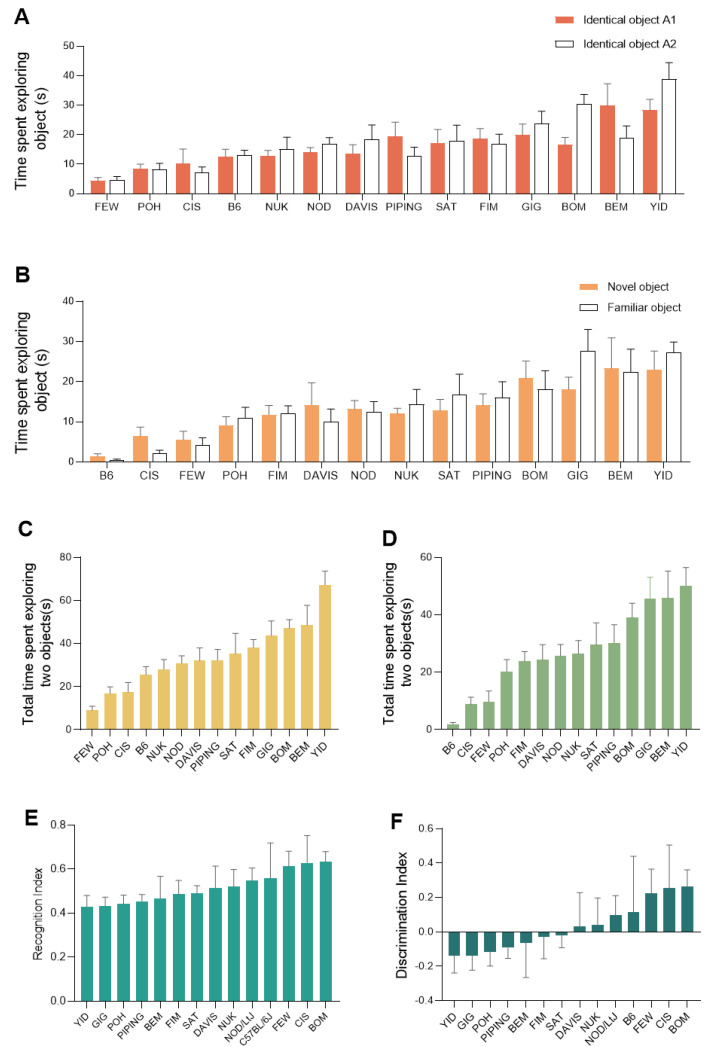
Phenotypic diversity in novel object recognition test traits among the CC population. (**A**) exploration time in the training phase (s). (**B**) Exploration time in the testing phase (s). (**C**) Total time spent exploring two objects in the training phase (s). (**D**) Total time exploring two objects in the testing phase (s). (**E**) recognition index. (**F**) Discrimination index. Plots are expressed as mean ± SEM, with CC lines ordered along the *x*-axis by mean per phenotype. ((**A**,**B**) Two-way repeated measures (mixed model) ANOVA test. (**A**) object x group interaction effect: *p* = 0.113, object: *p* = 0.234, group: *p* < 0.001; (**B**) object x group interaction effect: *p* = 0.540, object: *p* = 0.557, group: *p* < 0.001. (**C**−**F**) one-way ANOVA *p* < 0.001, *p* < 0.001, *p* = 0.249, and *p* = 0.249).

**Figure 4 ijms-24-00682-f004:**
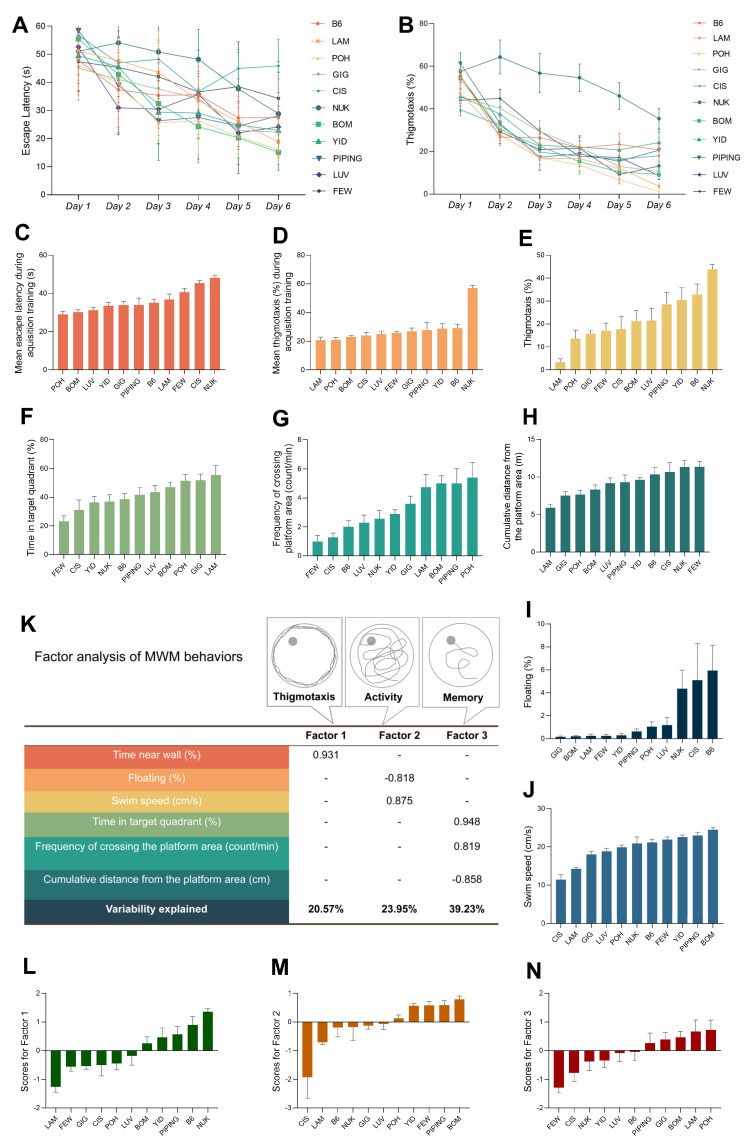
Phenotypic diversity in MWM task traits among the CC population. (**A**) Escape latency (s) to reach the platform on each day across the learning acquisition phase. (**B**) Thigmotaxis (%) to reach the platform on each day across the learning acquisition phase. (**C**) Mean escape latency (s) to reach the platform across the learning acquisition phase. (**D**) Mean Thigmotaxis (%) to reach the platform across the learning acquisition phase. (**E**) Thigmotaxis (%) in the target quadrant across the probe trial phase. (**F**) Time spent in the target quadrant across the probe trial phase (percent of total time, chance level = 25%). (**G**) Frequency of crossing platform area (count/min) across the probe trial phase. (**H**) Cumulative distance from the platform (m) to reach the platform across the probe trial phase. (**I**) Time spent floating (%) across the probe trial phase. (**J**) Average swim speed (m/s) across the probe trial phase. Plots are expressed as mean ± SEM, with CC lines ordered along the *x*-axis by mean per phenotype. (**K**) Factor analysis of behavioral variables during the probe trial phase of MWM task. Only factor loadings less than −0.5 or greater than 0.5 which represent strong correlations are shown in the table, whereas factor loadings between −0.5 and 0.5 are shown as hyphens. In the data set, factor analysis extracted three factors that together explain approximately 83.75% of the overall variance. Factor 1 (thigmotaxis), accounted for 20.57% of the variability, represents tendency to cling or follow the wall around the outer perimeter of the tank; factor 2 (activity), accounted for 23.95% of the variability, has strong negative factor loadings for floating and strong positive factor loadings for swim speed; factor 3 (memory), accounted for 39.23% of the variability, associated with more time in the target quadrant, less frequency of crossing the platform area, and less cumulative distance from the platform. (**L**–**N**) Mean factor score (±SEM) of each CC strain for the factors extracted from behavioral variables measured during the probe trial phase of MWM task. (**L**) Factor 1; (**M**) Factor 2; (**N**) Factor 3. Factor scores are displayed as deviation from zero in units of standard deviation, and the mean factor score of all 94 subjects are zero. ((**A**) Two-way repeated measures ANOVA test detected a strain x day interaction effect (*p* < 0.001), an effect of strain (*p* < 0.001) and an effect of day (*p* < 0.001) on the latency to find the platform during acquisition training. (**B**) Two-way repeated measures ANOVA test detected a strain × day interaction effect (*p* < 0.001), an effect of strain (*p* < 0.001) and an effect of day (*p* < 0.001) on the thigmotaxis during acquisition training. (**C**–**J**) one-way ANOVA *p* < 0.001. (**L**–**N**) one-way ANOVA *p* < 0.001).

**Figure 5 ijms-24-00682-f005:**
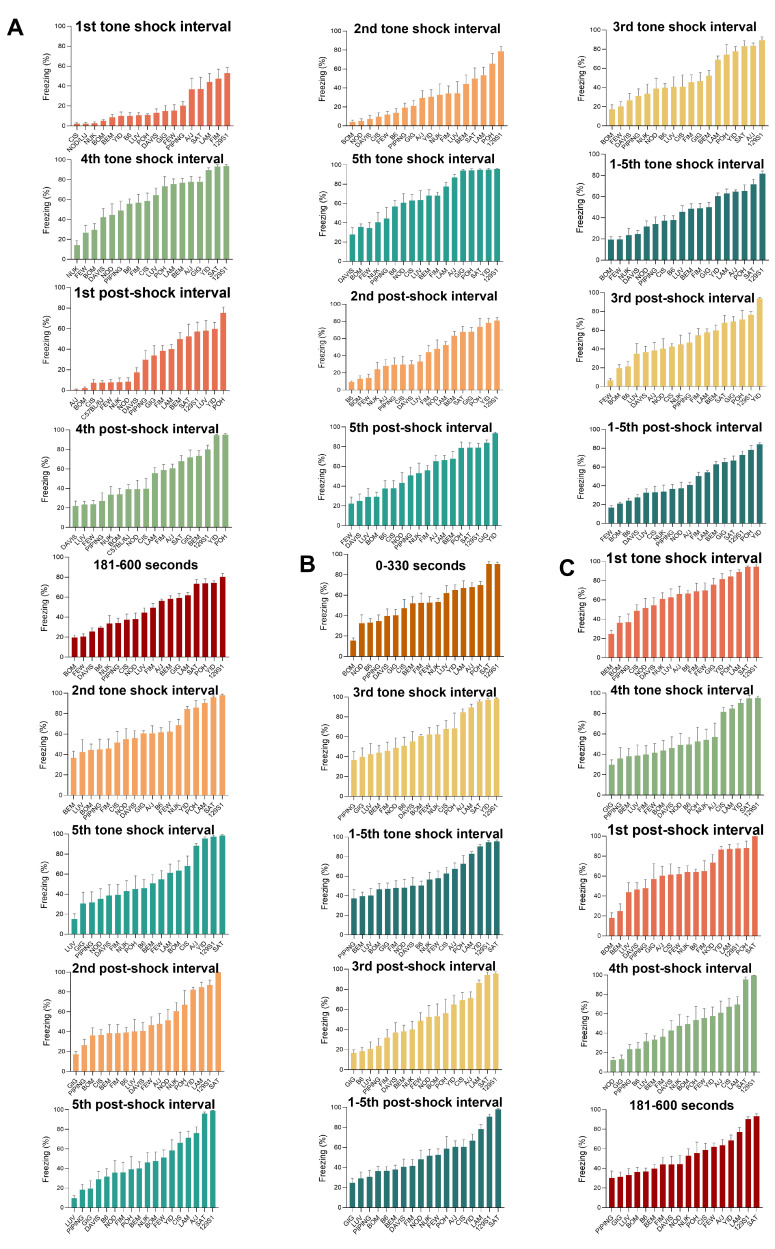
Phenotypic diversity in fear conditioning traits among the CC population. (**A**) 13 parameters used as indices of fear conditioning acquisition: the percentage of time spent freezing during the first, second, third, fourth, fifth tone shock interval, the percentage of time spent freezing during the total five tone intervals on Day 2 for fear conditioning training, the percentage of time spent freezing during the first, second, third, fourth and fifth post-shock interval, the percentage of time spent freezing during the total five post-shock intervals, the percentage of time spent freezing during 181–600 s on Day 2 for fear conditioning training, respectively. (**B**) A parameter used as an index of contextual fear memory: the percentage of time spent freezing during 0–330 s on Day 3 for contextual fear conditioning. (**C**) 13 parameters used as indices of cued fear memory: The percentage of time spent freezing during the first, second, third, fourth, fifth tone shock interval, the percentage of time spent freezing during the total five tone intervals on Day 4 for fear conditioning training, the percentage of time spent freezing during the first, second, third, fourth and fifth post-shock interval, the percentage of time spent freezing during the total five post-shock intervals, the percentage of time spent freezing during 181–600 s on Day 4 for cued fear conditioning, respectively. (Each parameter in (**A**–**C**) one-way ANOVA test *p* < 0.001).

**Figure 6 ijms-24-00682-f006:**
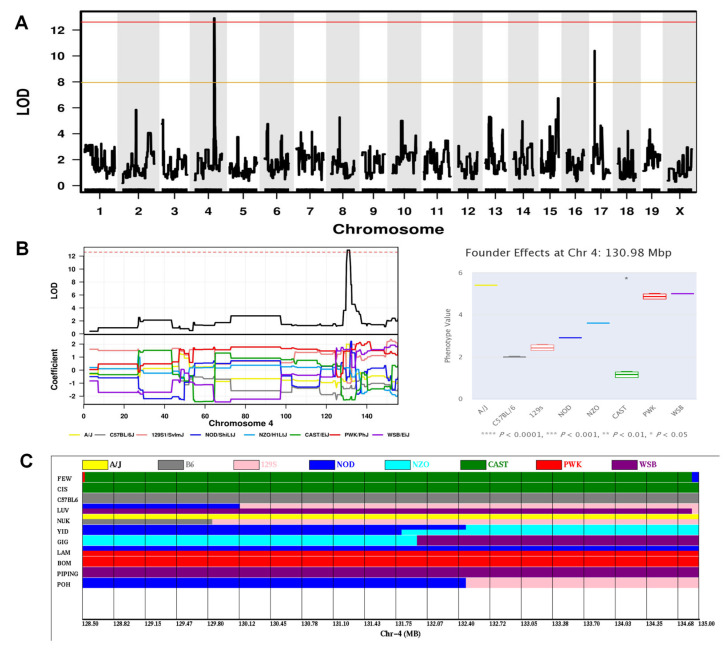
Mapping a significant QTL on Chr4 influencing the frequency of crossing the platform area on the probe trial phase of MWM. (**A**) Genome-wide LOD scores. The *x*-axis shows the chromosomal position and the *y*-axis shows the LOD scores. The solid red and orange horizontal lines indicate thresholds of 95% and 63% genome-wide significance, respectively. (**B**) LOD scores (upper left panel) on Chr4, and founder coefficient plot (bottom left panel) for Chr4. The dashed red horizontal line indicates thresholds of 95% significance. The association of founder haplotypes at 130.98 Mbp with this trait analyzed is shown in the right plot, indicating that the CAST haplotype significantly (*p* < 0.05) associated with lowest frequency of crossing the platform area. (**C**) Founder haplotypes in all CC strains at position 128.5–135.0 Mbp on Chr4. Strains are listed in the order from lower frequency (top) to higher frequency (bottom).

**Figure 7 ijms-24-00682-f007:**
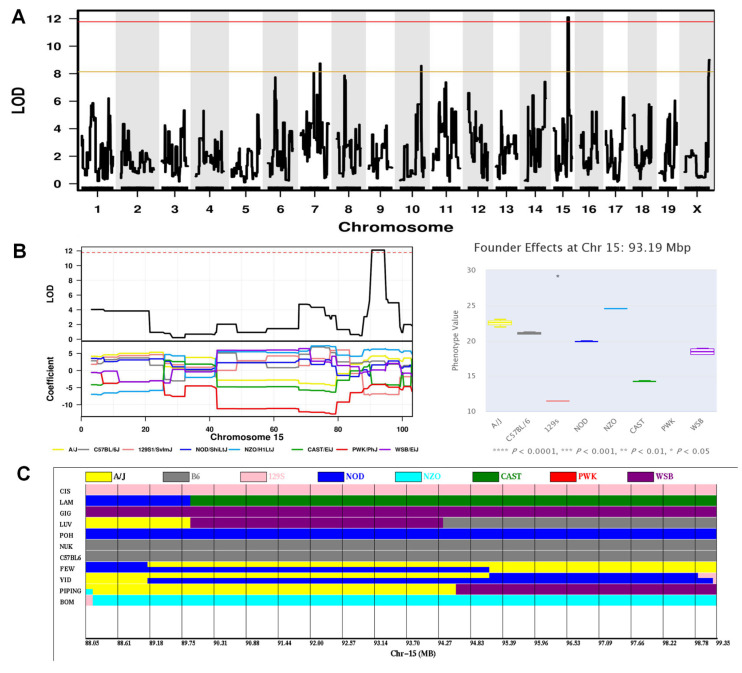
Mapping a significant QTL on Chr15 influencing the swim speed on the probe trial phase of MWM. (**A**) Genome-wide LOD scores. The *x*-axis shows the chromosomal position and the *y*-axis shows the LOD scores. The solid red and orange horizontal lines indicate thresholds of 95% and 63% genome-wide significance, respectively. (**B**) LOD scores (upper left panel) on Chr15, and founder coefficient plot (bottom left panel) for Chr15. The dashed red horizontal line indicates thresholds of 95% significance. The association of founder haplotypes at 93.19 Mbp with this trait analyzed is shown in the right plot, indicating that the haplotypes of CAST and 129S1 (*p* < 0.05) were associated with lowest swim speed. (**C**) Founder haplotypes in all CC strains at position 88.05–99.35 Mbp on Chr15. Strains are listed in the order from slower swim speed (top) to faster swim speed (bottom).

**Figure 8 ijms-24-00682-f008:**
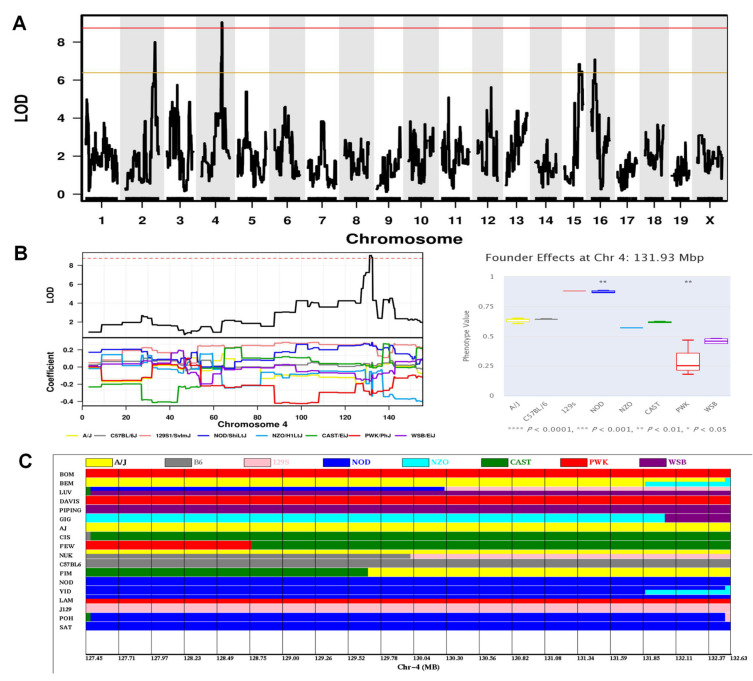
Mapping a significant QTL on Chr4 influencing the percentage of time spent freezing during the first post-shock interval on Day 4 for cued fear conditioning. (**A**) Genome-wide LOD scores. The *x*-axis shows the chromosomal position and the *y*-axis shows the LOD scores. The solid red and orange horizontal lines indicate thresholds of 95% and 63% genome-wide significance, respectively. (**B**) LOD scores (upper left panel) on Chr4, and founder coefficient plot (bottom left panel) for Chr4. The dashed red horizontal line indicates thresholds of 95% significance. The association of founder haplotypes at 131.93 Mbp with this trait analyzed is shown in the right plot, indicating that the NOD haplotype (*p* < 0.01) and 129S1 haplotype were associated with highest percentage of time spent freezing, while the PWK haplotype (*p* < 0.01) was associated with lowest percentage of time spent freezing. (**C**) Founder haplotypes in all CC strains at position 127.45–132.63 Mbp on Chr4. Strains are listed in the order from lower percentage (top) to higher percentage (bottom).

**Table 1 ijms-24-00682-t001:** Summary of significant behavior QTLs and best candidate genes.

QTL	Phenotype	Chr	Position (Interval, Mb)	Width (Mb)	Peak LOD	Sig. Level	Best Candidate Genes
*Sml2*	Frequency of crossing the platform area on the probe trial phase of MWM (spatial memory index)	4	128.5–135.0	6.50	12.9	0.05	*Lck, Ox1r, Sdc3, Ptafr, Srsf4, Stx12, Phactr4*
*Lal3*	Swim speed on the probe trial phase of MWM (locomotor activity index)	15	88.05–99.35	11.3	12.1	0.05	*Cntn1, Prickle1, Kif21a, Lrrk2, Kmt2b, Hdac7, Senp1,* *Plxnb2, Sbf1, Nell2*
*Cfml1*	The percentage of time spent freezing during the first post-shock interval on Day 4 for cued fear conditioning (cued fear memory index)	4	127.45–132.63	5.18	9.0	0.05	*Ak2, Hpca, Phactr4, Sdc3, Azin2, Hdac1*
*Lal4*	Time spent immobile in an open field during a 5 min test period in the open field test (locomotor activity index)	4	149.98–152.61	2.63	9.8	0.1	*Tnfrsf9, Per3, Camta1*
*All4*	Total distance traveled in the periphery of an open field during a 5 min test period in the open field test (anxiety level index)	7	149.55–152.51	2.96	8.3	0.1	*Shank2*, *Ctsd*, Igf2, Th
*Cfml2*	The percentage of time spent freezing during the third post-shock interval on Day 4 for cued fear conditioning & The percentage of time spent freezing during the fifth post-shock interval on Day 4 for cued fear conditioning & The percentage of time spent freezing during five post-shock intervals on Day 4 for cued fear conditioning(cued fear memory index)	2	156.60–170.90	14.3	8.8	0.1	*Src3, Neurl2, Elmo2, Sulf2, Pigt, Ctnnbl1, Slc12a5, Cd40*

## Data Availability

All data needed to evaluate the conclusions in the paper are presented in the paper and the Appendix A. Additional data are available from the corresponding author on reasonable request.

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
