# Peer review of "Genetic Mapping of Behavioral Traits Using the Collaborative Cross Resource"

_ijms, 2022, doi:10.3390/ijms24010682_

Round 1

Reviewer 1 Report

The paper by Xuan et al. is an interesting attempt to advance the knowledge of the genetics of behavioral traits using the collaborative cross (CC) mouse resource. Their study highlights the potential of the CC population in behavioral genetic research because they provide the variations needed for studying quantitative traits.

They report assessing 52 behavioral measures associated with locomotor activity, anxiety level, learning, and memory. The results presented substantial continuous behavioral variations across the CC strains tested. These results enabled mapping six quantitative trait loci (QTLs) that influenced these traits.

The analysis of the founder haplotypes of the CC strains at the positions of the highest QTL scores, followed by a search of the Sanger database to identify the candidate genes of the founder possessing founder-specific protein-changing single-nucleotide polymorphisms  (SNPs) or insertions-deletions (Indels) was nicely used to define the candidate genetic variants underlying these QTLs.

The suggestion of the genes affecting behavioral traits is the paper's main contribution, and they are nicely discussed in the discussion session.

Minor comments:

1. The use of : test period (s), s should be written as sec, which defines second or the s explained as second.

2. Consider adding the number of mice used for each behavioral test in each figure, possibly under the name of the strain (Figures 2,3,4,5.

3. Figure 5 indeed does show the continuous pattern, but the details are completely obscured. Consider enlarging the figure. Additionally, for figure 5, what does the “respectively” stand for? Please rephrase.

Reviewer 2 Report

1. It is suggested to label significant difference in the graphs (if appliable).

2. It is suggested to provide the age and weight information of mice. 

Reviewer 3 Report

Dear authors

This is a very well written paper and it concerns a very interesting subject. I would like to elaborate more on the albino lines and discuss how their genetic background is related with the changes observed. Also, minor changes in english are required.
